# MMNMT: Modularizing Multilingual Neural Machine Translation with Flexibly Assembled MoE and Dense Blocks

**Shangjie Li**
Tianjin University
sj_li@tju.edu.cn

**Xiangpeng Wei**
pemywei@gmail.com

**Shaolin Zhu**
Tianjin University
zhushaolin@tju.edu.cn

**Jun Xie**
stiffxj@gmail.com

**Baosong Yang**[*]
nlp2ct.baosong@gmail.com

**Deyi Xiong**[*]
Tianjin University
dyxiong@tju.edu.cn

## Abstract

Mixture-of-Experts (MoE) based sparse architectures can significantly increase model capacity with sublinear computational overhead, which are hence widely used in massively multilingual neural machine translation (MNMT). However, they are prone to overfitting on low-resource language translation. In this paper, we propose a modularized MNMT framework that is able to flexibly assemble dense and MoE-based sparse modules to achieve the best of both worlds. The training strategy of the modularized MNMT framework consists of three stages: (1) Pre-training basic MNMT models with different training objectives or model structures, (2) Initializing modules of the framework with pre-trained couterparts (e.g., encoder, decoder and embedding layers) from the basic models and (3) Fine-tuning the modularized MNMT framework to fit modules from different models together. We pre-train three basic MNMT models from scratch: a dense model, an MoE-based sparse model and a new MoE model, termed as MoE-LGR that explores multiple Language-Group-specifc Routers to incorporate language group knowledge into MNMT. The strengths of these pre-trained models are either on low-resource language translation, high-resource language translation or zero-shot translation. Our modularized MNMT framework attempts to incorporate these advantages into a single model with reasonable initialization and fine-tuning. Experiments on widely-used benchmark datasets demonstrate that the proposed modularized MNMT framwork substantially outperforms both MoE and dense models on high- and low-resource language translation as well as zero-shot translation. Our framework facilitates the combination of different methods with their own strengths and recycling off-the-shelf models for multilingual neural machine translation. Codes are available at https://github.com/lishangjie1/MMNMT.

## 1 Introduction

Multilingual neural machine translation translates multiple languages within a single model via multi-task learning, facilitating the deployment of machine translation service in practice and improving low/zero-resource language translation (Johnson et al., 2017; Aharoni et al., 2019; Zhang et al., 2020; Fan et al., 2021; Team et al., 2022). However, as the number of languages translated by an MNMT model increase, the capacity of the model has to be increased accordingly, otherwise translation quality will degrade for all language pairs, especially for high-resource languages. This phenomenon is referred to as capacity bottleneck (Aharoni et al., 2019; Zhang et al., 2020). MoE models provide an effective way to increase model capacity while the computation cost is sublinear to the number of parameters. Due to this advantage of MoE models over dense models, sparse architectures built on MoE models are widely explored for massively multilingual neural machine translation that requires large model capacity. However, MoE models suffer from overfitting on low-resource languages (Team et al., 2022), which is not observed in MNMT built on dense models.

This inspires us to ask a question: can we achieve the best of both worlds of dense and MoE-based sparse architectures for multilingual NMT? To answer this question, we propose MMNMT that Modularizes Multilingual NMT with flexibly assembled dense and MoE blocks. Specifically, the training strategy of MMNMT consists of three stages:

- Pre-training basic multilingual NMT models with different architectures, attempting to explore the strengths of different models on low-resource language translation (e.g., dense models), high-resource language translation (e.g., MoE models).

- Initializing the modules of the proposed mod-

---

[*]Corresponding authors.

ularized MNMT model with blocks from the pre-trained basic multilingual NMT models, e.g., using the pre-trained dense encoder to initialize the encoder of MMNMT. Such initialization can be done in a flexible module assembling way.

- Fine-tuning the MMNMT model to make the assembled modules fit together.

Using dense modules to initialize MoE models in our training strategy is in line with our preliminary experiments and recent studies on MoE-based language models. These studies find that training MoE-based language models from an off-the-shelf dense model (e.g., T5 (Raffel et al., 2020)) is more efficient than training from scratch (Nie et al., 2022; Komatsuzaki et al., 2023). Our preliminary experiments also demonstrate that dense MNMT models are superior to MoE-based sparse models on low-resource language translation.

In order to diversify the options of pre-trained basic models and improve zero-shot translation in MMNMT, we further propose MoE-LGR that incorporates a Language Group Router into MoE models. Routing mechanism plays an important role in token dispatch and resource allocation for MoE models. Previous works in this line mainly focus on load balancing across experts in MoE (Lewis et al., 2021; Roller et al., 2021; Fedus et al., 2022; Zuo et al., 2022; Zhou et al., 2022) to prevent experts from being not specialized or overly specialized. However, the language information of each token is not fully explored in routers, which could enhance the generation ability of the decoder and mitigate the off-target problem in zero-shot translation. To address this issue, we introduce MoE-LGR. Particularly, we categorize languages into multiple groups according to linguistic typology and language embedding clustering, and learn a router per language group to enhance the difference of routing for different language groups.

The main contributions of this work are summarized as follows:

- We propose MMNMT to modularize multilingual NMT, which is capable of assembling modules from both dense and sparse models and fitting them together to achieve the best of both worlds.

- We present MoE-LGR with language group routers, which is able to significantly improve zero-shot translation in multilingual NMT.

- Experiments on the OPUS-100 and PC32 dataset demonstrate that the proposed MMNMT achieves significant improvements over both MoE and dense models on all language directions, especially on low-resource and zero-shot translation.

## 2 Related Work

**Multilingual Neural Machine Translation** Multilingual neural machine translation has been gaining increasing interest in recent years (Johnson et al., 2017; Aharoni et al., 2019; Zhang et al., 2020; Fan et al., 2021; Team et al., 2022). However, MNMT models tend to be inferior to bilingual NMT counterparts on translating high-resource languages due to the capacity bottleneck (Zhang et al., 2020). A wide variety of approaches have been proposed to alleviate this issue, e.g., deepening models to increase model capacity (Zhang et al., 2020), exploring lightweight language-specific modules (Philip et al., 2020; Zhang et al., 2021; Zhu et al., 2021), word alignments (Lin et al., 2020), contrastive learning (Pan et al., 2021) and Mixture-of-Experts (Lepikhin et al., 2020; Fedus et al., 2022; Nie et al., 2022). Mixture-of-Experts based MNMT models usually suffer from overfitting problem (Team et al., 2022). Alleviating the overfitting problem is a key focus of our proposed modularized MNMT framework.

**Mixture-of-Experts** Sparsely-gated Mixture-of-Experts (MoE) (Lepikhin et al., 2020; Fedus et al., 2022) selects the top-K experts in MoE layers through a routing mechanism. A strand of research focuses on load balancing, such as Base Layer (Lewis et al., 2021), Hash Layer (Roller et al., 2021), THOR (Zuo et al., 2022), expert choice routing (Zhou et al., 2022). In this work, we propose a language group routing mechanism to provide language information and enhance the difference of routing across language groups, which significantly improves the performance of zero-shot translation. Recently, Komatsuzaki et al. (2023) use a pre-trained T5 dense model to initialize an MoE language model, achieving improvements in the computational cost but no obvious performance gains. Our work explores modularized initialization of different parts of MoE models with various pre-trained basic models for multilingual NMT, which is flexible and effective.

| Model | Any2En | | | | En2Any | | | |
|---|---|---|---|---|---|---|---|---|
| | High | Medium | Low | Avg | High | Medium | Low | Avg |
| Dense | 29.6 | 40.4 | **33.5** | 32.3 | 22.5 | 33.2 | 29.7 | 26.3 |
| MoE | **32.6** | **41.9** | 32.3 | **33.7** | **25.5** | **37.9** | **33.0** | **29.6** |

Table 1: BLEU scores of the dense vs. MoE model on both Any-to-English and English-to-Any translation.

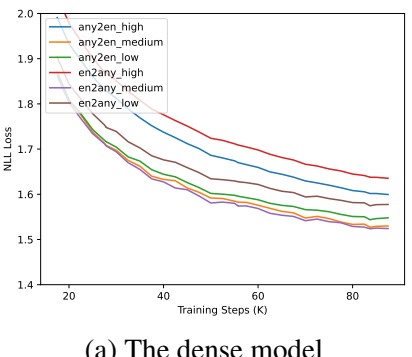

(a) The dense model

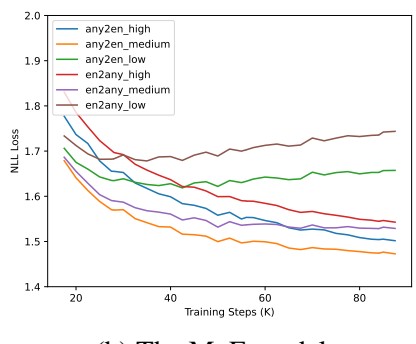

(b) The MoE model

Figure 1: The validation loss of the dense model and MoE model during training.

## 3 Preliminary Experiments and Findings

To have a deep understanding on the strengths and weaknesses of dense and MoE models in multilingual NMT, we conducted preliminary experiments to compare them. Details of these experiments can be found in section 5.1. Results are shown in Table 1. We observe that the MoE model significantly outperforms the dense model on high-resource language translation, and achieves consistent improvements over the dense model on all En-to-Any directions. This demonstrates the advantage of MoE sparse models over dense models in terms of model capacity. However, we also find that MoE models are prone to overfitting on low-resource language translation, which resonates with the finding of (Team et al., 2022). In Any-to-English translation, the dense model outperforms the MoE model by 1.3 BLEU (33.5 vs. 32.3) on low-resource language tanslation. In addition to this, we plot the validation loss of the dense model and MoE model during training in Figure 1. It is obvious that the validation losses of both Any-to-English and English-to-Any low-resource language translation initially decrease and later increase, confirming the existence of the overfitting problem.

## 4 Methodology

To mitigate the overfitting issue and achieve the best of both worlds, we propose a general framework MMNMT to modularize multilingual NMT so as to assemble desirable modules from both MoE and dense models. The training strategy of MMNMT consists of basic model pre-training, module initializing and fine-tuning.

### 4.1 Basic Model Pre-training

We introduce three types of basic models for our general framework, namely dense model, MoE model and the proposed MoE-LGR model. These basic models are trained from scratch with cross-entropy objectives on training data.

**Dense Model** The dense model is a encoder-decoder backbone network contains 12 Transformer encoder blocks and 12 Transformer decoder blocks. The encoder and decoder have a shared embedding layer.

**MoE Model** The MoE model substitutes the feed-forward network (FFN) sublayer in the Dense model with an MoE layer that consists of multiple FFN experts $\{FFN_i\}_{i=1}^{N}$ to expand model capacity. The MoE model uses a token router $p$ (typically a Top-K gate) to perform token dispatch to experts.

**MoE-LGR** To explore language information and improve zero-shot translation capability in the decoder, we add multiple language-group-specific routers (one router per language group) into MoE layers of the decoder to enhance the difference of routing across language groups.

The inputs to the language group router are a token representation $\mathbf{x}$ and a language ID lid. A language group will be automatically identified

according the language ID via the language group identification operation LGI. The shared router and the identified language-group-specific router is aggregated to yield the output of the language group router, which is computed as follows:

$$p(\mathbf{x}) = \text{TopK}(\text{softmax}(\frac{\mathbf{W}_{\text{share}}\mathbf{x} + \mathbf{W}_{\text{LGI(lid)}}\mathbf{x}}{2})) \tag{1}$$

where $\mathbf{W}_{\text{share}}$ is the weight of share router across all languages, $\mathbf{W}_{\text{LGI(lid)}}$ is the weight of the language-group-specific router identified by LGI(lid).

For language groups, we use both external linguistic typology knowledge and internal language token embeddings to group languages. First, we categorize languages into multiple groups following external linguistic typology from Ethnologue (Lewis et al., 2009), which is one of the most authoritative language family taxonomy. Specifically, as we use the OPUS-100 (Zhang et al., 2020) dataset in our experiments, which is English-centric and contains 99 language pairs (as the test sets in OPUS-100 only cover 94 language pairs, we only use 94 language pairs in supervised training), we divide 95 languages (including English) into 22 groups according to Ethnologue. The majority of languages are from the Indo-European family. Some languages are assigned to a group with only one language of their own. For example, Thai is from the Tai-Kadai group and no other languages in the OPUS-100 dataset are assigned to this group.

In order to balance the corpus size across groups, we need to further restructure language groups according to language similarity. Language embedding based clustering (Tan et al., 2019) is a solution to automatically cluster languages into groups by using language token embeddings learned by MNMT. However, this method may generate low-quality language clusters due to low quality of low-resource language embeddings. Hence, we cluster languages on the basis of linguistic typology (i.e., based on 22 groups defined in Ethnologue). We calculate group embeddings as follows:

$$\mathbf{E}_{\text{G}} = \sum_{i \in \text{G}} \frac{\text{T}_i}{\text{T}_{\text{G}}} \mathbf{E}_i \tag{2}$$

where $\mathbf{E}_{\text{G}}$ is the group embedding, $\mathbf{E}_i$ is the language embedding of language i in group G, $\text{T}_i$ is the corpus size of the language i, $\text{T}_{\text{G}}$ is the total corpus size of all languages in group G.

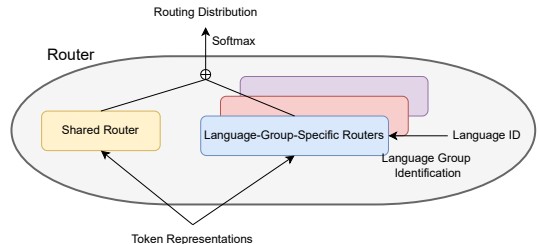

Figure 2: Diagram of the proposed language group router.

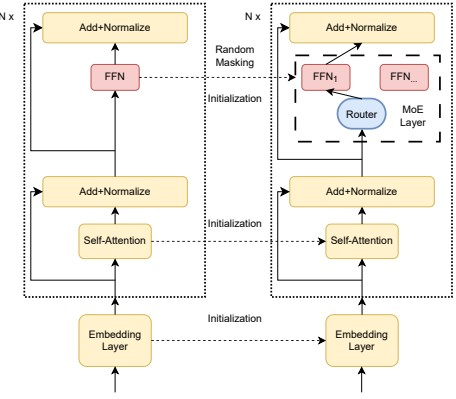

Figure 3: Illustration of the initialization from the encoder of the pre-trained dense model to the encoder of MMNMT. The decoder is initialized in the same way, which is omitted here.

In this way, the impact of the low-quality language embeddings of low-resource languages is reduced. Then, we perform hierarchical clustering with computed group embeddings and set a maximum group corpus size $\text{T}_{\text{max}}$ to prevent excessive clustering. Specifically, let M the number of the initial groups after the linguistic typology knowledge based grouping. In each iteration, we merge the two closest groups a and b whose total corpus size is less than $\text{T}_{\text{max}}$, removing the two old groups and forming a new group. After this, the number of groups decreases from M to M-1. The group embedding of the new group is obtained by weighting the old group embedding based on the corpus size:

$$\mathbf{E}_{\text{new}} = \frac{\text{T}_{\text{a}} * \mathbf{E}_{\text{a}} + \text{T}_{\text{b}} * \mathbf{E}_{\text{b}}}{\text{T}_{\text{a}} + \text{T}_{\text{b}}} \tag{3}$$

When grouping no longer changes, we obtain the final grouping result. In this way, we obtain 4 groups for OPUS-100 dataset at last. Please refer to Appendix A for more details.

| ID | Wall Time (Hour) | Model | Any2En | | | | En2Any | | | | Zero-Shot | |
|---|---|---|---|---|---|---|---|---|---|---|---|---|
| | | | High | Medium | Low | Avg | High | Medium | Low | Avg | BLEU | LangAcc(%) |
| | | | | | | Basic Models | | | | | | |
| (1) | 28 | Dense | 29.6 | 40.4 | 33.5 | 32.3 | 22.5 | 33.2 | 29.7 | 26.3 | 4.0 | 16.2 |
| (2) | 79 | Dense | 30.1 | 40.9 | 33.5 | 32.7 | 23.1 | 33.9 | 30.4 | 26.9 | 4.2 | 16.5 |
| (3) | 79 | MoE | 32.6 | 41.9 | 32.3 | 33.7 | 25.5 | 37.9 | 33.0 | 29.6 | 5.4 | 24.7 |
| (4) | 158 | MoE | 32.9 | 42.5 | 32.5 | 34.1 | 25.8 | 38.5 | 33.5 | 30.0 | 5.3 | 24.0 |
| (5) | 79 | MoE-LGR | 32.5 | 41.7 | 31.9 | 33.5 | 25.5 | 38.0 | 32.8 | 29.6 | 7.1 | 32.9 |
| (6) | 158 | MoE-LGR | 33.0 | 42.3 | 32.3 | 34.0 | 25.9 | 38.5 | 33.4 | 30.0 | 7.2 | 34.2 |
| | | | | | | MMNMT Models | | | | | | |
| (7) | 28+79 | $Enc_{(1)} - Dec_{rand}$ | 32.8 | 43.3 | **35.1** | 34.9 | 25.5 | 37.9 | 33.6 | 29.9 | 4.6 | 17.2 |
| (8) | 28+130 | $Enc_{(1)} - Dec_{rand}$ | 33.0 | **43.6** | 34.9 | **35.0** | 25.6 | 38.5 | 33.9 | 30.1 | 4.7 | 20.1 |
| (9) | 28+79 | $Enc_{(1)} - Dec_{(1)}$ | 32.7 | 42.9 | 34.3 | 34.5 | 25.3 | 38.0 | 34.4 | 30.0 | 5.9 | 27.6 |
| (10) | 28+130 | $Enc_{(1)} - Dec_{(1)}$ | 32.9 | 43.1 | 34.2 | 34.6 | 25.6 | 38.1 | **34.6** | 30.2 | 6.2 | 29.8 |
| (11) | 79+79 | $Enc_{(1)} - Dec_{(3)}$ | 33.3 | 42.1 | 32.2 | 34.0 | 26.2 | 38.7 | 33.2 | 30.2 | 7.0 | 32.6 |
| (12) | 79+79 | $Enc_{(1)} - Dec_{(5)}$ | **33.3** | 42.4 | 32.9 | 34.3 | **26.4** | **38.8** | 33.2 | **30.3** | 9.6 | **44.9** |

Table 2: Results of different models on the OPUS-100 test set. A subscript indicator denotes that the corresponding module of the MMNMT model is initialized by the counterpart of the basic model indexed by the indicator. LangACC: the proportion of translations in the correct target language among all translations.

## 4.2 Module Initialization

Once basic models are pre-trained, we can use parameters of these pre-trained models to initialize modules of the general MMNMT framework or the entire framework in different ways.

**Initialization From Dense Model** When we use the pre-trained dense model to initialize MMNMT, we perform initialization in a layer-wise way. As shown in Figure 3, we initialize the parameters of MMNMT layers with those of the dense model layers that have the same structure as the corresponding MMNMT layers, e.g., self-attention layer, cross-attention layer, embedding layer. For FFN experts initialization, we randomly masks parameters of the corresponding FFN in the dense model and choose to initialize an FFN expert from MMNMT in the same layer with the remaining parameters. In doing so, we can enhance the diversity of experts (Nie et al., 2022).

**Initialization from MoE Model** This can be done in a straightforward way as they have the same architecture. The initialization is performed for the entire model.

**Initialization from MoE-LGR Model** As the MoE-LGR has multiple language-group-specific routers in the decoder, we need to add multiple language-group-specific routers in the decoder of MMNMT accordingly to accommodate the initialization.

**Mixed Initialization** To combine the strengths of these basic models, we can perform mixed initialization. For example, we can use the encoder of the dense model and the decoder of the MoE-LGR model to initialize the MMNMT model, which could improve the ability of low-resource language translation as well as zero-shot translation.

## 4.3 Fine-tuning

After initializing the MMNMT model, we need to further fine-tune the entire model on the same training data used in the basic model pre-training to ensure that the various modules in the model cooperate with each other, especially under the mixed initialization condition. We fine-tune the MMNMT model with cross-entropy and load-balance objectives.

## 5 Experiments

We used the MoE branch of fairseq[1] to implement our MoE models. We conducted extensive experiments to examine the effectiveness of the proposed MMNMT against MoE and dense model baselines in multilingual machine translation.

## 5.1 Experiment Setting for Many-to-Many Translation

**Dataset** We used the publicly available OPUS-100 dataset which contains approximately 55M sentence pairs and 99 language pairs in our experiments. Since the test sets in OPUS-100 only covers 94 language pairs, we only used the data of these 94 language pairs for training. We employed the temperature-based sampling strategy (Aharoni et al., 2019) with T = 5 to balance corpus size for various language pairs. We used SentencePiece

---

[1]https://github.com/facebookresearch/fairseq/tree/moe

([Kudo and Richardson](#), 2018) for tokenization and the vocabulary size was 64K.

**Model Configurations**  Our dense MNMT model contained 12 Transformer encoder blocks and 12 Transformer decoder blocks, where the model dimension was 1024, the number of attention heads was 8. For the MoE model, MoE layer frequency of the MoE MNMT model was 2, the capacity factor was 1.25 in training and the eval-capacity-factor was set to 0.75 in inference. Every MoE layer had 32 experts and Top-2 routing strategy was used for expert routing. Other configurations were the same as the dense model. For the MoE-LGR model, we set $T_{max} = 15M$ and used the language embeddings learned by the dense model. The 95 languages were finally clustered into 4 language groups.

**Training Configurations**  We conducted all experiments on 8 NVIDIA Tesla V100 32GB GPUs. For basic model pre-training and MMNMT fine-tuning, we used the Adam optimizer ([Kingma and Ba](#), 2014) with learning rate = $2e^{-4}$, $\alpha = 0.9, \beta = 0.98$. The learning schedule was Polynomial decay with the number of warm-up steps being set to 4000, the end learning rate was $1e^{-5}$, and the total number of updates was 100K. We set the training max tokens to 4096 per GPU and accumulated the gradients every 4 steps. The training objects were cross-entropy loss and load-balance loss with a weight of 0.1. The best checkpoint was selected according to the perplexity (ppl) on the validation set. The BLEU score was computed via sacrebleu ([Post](#), 2018), and we employed langdetect[2] toolkit for language identification on the OPUS-100 dataset to calculate language accuracy of target translations.

## 5.2  Main Results

In order to examine the effectiveness of different models, we evaluated the translation performance on the OPUS-100 test set and OPUS-100 zero-shot test set. We grouped languages into three categories according to the size of training data available for them in the training dataset: High (500K$\leq$ size $\leq$ 1M, 100 directions), Medium (200K $<$ size $<$ 500K, 24 directions) and Low ($\leq$ 200K, 64 directions). The zero-shot test set covers six languages (Arabic, Chinese, Dutch, French, German, and Russian) and 30 zero-shot directions.

---

[2]https://github.com/Mimino666/langdetect

Results are shown in Table 2. The Wall Time is the training cost of models. The dense model and MoE model require 28 and 79 hours to complete 100K training steps respectively. For a fair comparison, we also provide results for a dense model that continues to train in 79 hours with 260K steps. The wall time of MMNMT model is in x+y format, x denotes the pre-training time of the corresponding basic model and y is the time of fine-tuning. We also extend the training time of some MMNMT models to conduct fair comparisons in terms of training hours. It can be seen that the MoE-LGR model is on a par with the MoE baseline on Any-to-En and En-to-Any translation, but achieves substantial improvements of 1.9 BLEU and 10.2% LangAcc over the MoE model on zero-shot translation (see (4) and (5) in Table 2). This is in line with our motivation in developing the MoE-LGR model to improve zero-shot translation via language-group-specific routers. All MMNMT models with different assembled modules are superior to both dense and MoE models. The MMNMT model $\text{Enc}_{(1)} - \text{Dec}_{\text{rand}}$, where the encoder is initialized with the encoder of the dense model and the decoder is randomly initialized, achieves an average improvement of 0.9 BLEU over the MoE model (see (4) and (8) in Table 2). The initialization from the encoder of the dense model significantly improves low-resource language translation by 2.4 BLEU on Any-to-En translation over the MoE model and is on a par with it on En-to-Any translation. $\text{Enc}_{(1)} - \text{Dec}_{(1)}$ which uses the pretrained dense model to initialize both encoder and decoder, outperforms the MoE baseline by an average of 0.5 BLEU on Any-to-English translation and 0.2 BLEU on English-to-Any translation. Especially, $\text{Enc}_{(1)} - \text{Dec}_{(1)}$ achieves 34.6 BLEU on English-to-Any low-resource language translation. The $\text{Enc}_{(1)} - \text{Dec}_{(3)}$ and $\text{Enc}_{(1)} - \text{Dec}_{(5)}$ model achieve significant improvements on high-resource language translation. Specifically, $\text{Enc}_{(1)} - \text{Dec}_{(5)}$ outperforms the MoE baseline by 0.4 and 0.6 BLEU on Any-to-English and English-to-Any high-resource language translation. These results suggest that the proposed MMNMT initialized with blocks from basic models is able to benefit from the advantages of these basic models.

## 5.3  Zero-Shot Results

Zero-shot translation enabled by multilingual NMT usually suffers from the off-target translation issue

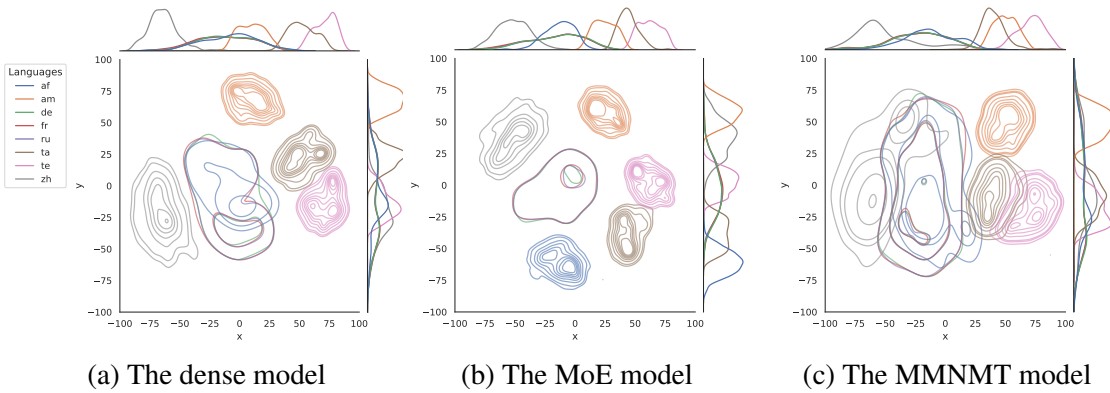

| (a) The dense model | (b) The MoE model | (c) The MMNMT model |

Figure 4: Visualization of the encoder top layer representations of different models.

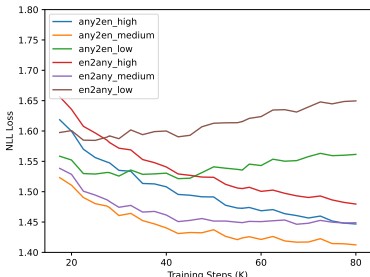

Figure 5: The validation loss of the MMNMT model during training.

(Zhang et al., 2020), where tokens in the target translation are not in the right language. We evaluated baselines and our methods on OPUS-100 zero-shot test set (Zhang et al., 2020). Results are shown in Table 2. The MoE model achieves 5.3 BLEU and 24.0% language accuracy, slightly mitigating the off-target problem compared with the dense model. The MoE-LGR model improves BLEU and language accuracy from 5.3 to 7.2 and 24.0% to 34.2% respectively. The language group router encourages similar token routing behavior within a language group. This makes the token-to-experts assignment in the decoder more language-specific, therefore improving the accuracy of language generation. The $\mathrm{Enc}_{(1)} - \mathrm{Dec}_{(5)}$ model achieves the best result on zero-shot translation, improving the BLEU score from 5.3 to 9.6 and language accuracy from 24.0% to 44.9%. This model initializes its decoder with the decoder of MoE-LGR, which itself achieves good performance on zero-shot translation.

## 5.4 Analysis on the Encoder Representations and Validation Loss Curve

In order to take a deep look into the improvements obtained by the initialization from the dense model,

we visualize the representations of the top layer of the encoder in Figure 4. We used the Flores-200 (Team et al., 2022) dataset, which is a many-to-many multilingual benchmark including 204 languages. We selected 4 high-resource languages (German, French, Russian and Chinese) and 4 low-resource languages (Afrikaans, Amharic, Tamil and Telugu) in Flores-200, and encoded all sentences by the encoder of three different models: (a) the dense model (b) the MoE model trained from scratch and (c) the MMNMT (Dense) model (i.e., $\mathrm{Enc}_{(1)} - \mathrm{Dec}_{\mathrm{rand}}$ in Table 2). We averaged the sequential representations of the sentences from the above 8 languages over the sequence dimension, and applied the t-SNE (Laurens and Hinton, 2008) dimensionality reduction algorithm to reduce the 1024 dimensions to two dimensions. Then we ploted the bivariate kernel density estimation based on the reduced 2-dim representations. According to Figure 4, the encoder top layer representations of the dense model are more compact than those of the MoE model. Russian representations seem to overlap with those of German, French, and Afrikaans. The MMNMT (Dense) model whose encoder is initialized by the encoder of the dense model, shows obviously better unified representations than the other two models. All 8 languages are close to each other in the shared space and gradually merged into a cluster.

We also plot the validation loss curve of the MMNMT (Dense) model during training in Figure 5. We observe an overall decrease in the validation loss of the MMNMT model compared with that of the MoE model ((b) in Figure 1), especially on low-resource language translation. This explains the significant improvements gained by the MMNMT model on low-resource language translation.

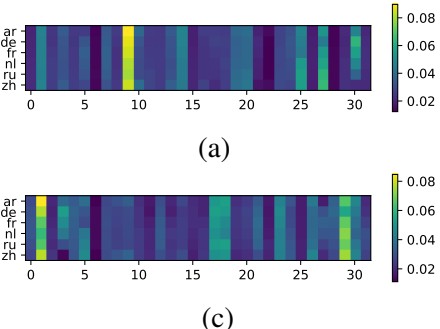 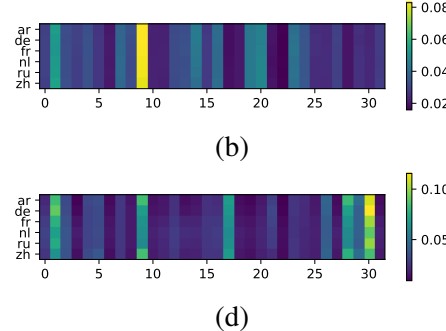

Figure 6: The token-to-experts assignments of the last MoE layer of the decoder for the MoE model and the MMNMT (MoE-LGR) model. (a): zero-shot translation by the MoE model. (b): X-to-English translation by the MoE model. (c): zero-shot translation by the MMNMT model. (d): X-to-English translation by the MMNMT model. Brightness represents the probability of experts being routed.

## 5.5 Analysis on the Token-to-Experts Assignments in the Decoder

To analyse the improvements of the MMNMT (MoE-LGR) model (i.e., $\text{Enc}_{(1)} - \text{Dec}_{(5)}$ in Table 2) gained on zero-shot translation, we report detailed results of zero-shot translation in Table 4. Compared with MoE baselines, the MMNMT (MoE-LGR) model improves BLEU scores by an average of 4.2 and language accuracy by an average of 20.3%. Arabic, Russian, and Chinese have the most noticeable improvements, which may be due to the fact that these languages are not categorized into the same group as English, making them more differentiated in routing and effectively alleviating the off-target issue. We visualize the token-to-experts assignments of zero-shot translation in the last MoE-layer of the decoder for MoE and MMNMT (MoE-LGR) respectively in Figure 6, where the x-axis represents 32 experts, and the y-axis denotes the target language (e.g., the first row in (a) is the average token-to-experts assignments of De-Ar, Fr-Ar, Nl-Ar, Ru-Ar, Zh-Ar). As zero-shot translation usually suffers from the off-target issue (i.e., easily translated into English), we used the same source sentences to translate them into English to observe changes in token-to-experts assignments. We observe that the MoE model has a similar token-to-experts assignments on zero-shot and X-to-English translation (see (a) and (b) in Figure 6), while MMNMT learns different assignments (see (c) and (d) in Figure 6). Our proposed language-group-specific routers route tokens from different language groups to different experts, mitigating the off-target issue of zero-shot translation.

## 5.6 Adapting Off-the-Shelf Dense Checkpoints to Sparse Architectures Via MMNMT

A wide variety of models have been proposed to improve dense multilingual machine translation in the past few years (Johnson et al., 2017; Aharoni et al., 2019; Zhang et al., 2020; Lin et al., 2020; Pan et al., 2021). Our framework facilitates the adaptation of these off-the-shelf dense checkpoints to MoE-based sparse architectures, the current dominant formalism for massively multilingual NMT. We consider public MNMT model checkpoints with available multilingual parallel training data. For this, we choose mRASP2 (Pan et al., 2021) for our dense-to-sparse adaptation experiments with off-the-shelf models.

**Experiment Settings** We used PC32 dataset (Lin et al., 2020) which is used by mRASP2. PC32 is a English-centric multilingual parallel corpus which includes 32 language pairs and 64 translation directions in total. The details of the dataset, model and training configuration can be found in Appendix B.

**Results** Results are shown in Table 3. We compared with five baselines, and the numbers in bracket are the number of model parameters and activated parameters at inference time. The dense and MoE models were trained on the PC32 dataset from scratch, and their model configuration were the same as mRASP2. mRASP2 (Pan et al., 2021), M2M-100 (Fan et al., 2021) and NLLB (Team et al., 2022) are all publicly released MNMT dense checkpoints, which were downloaded and evaluated without any changes. We applied our framework to both dense and mRASP2 models to obtain MMNMT (Dense) and MMNMT (mRASP2) re-

| Model (#params / #activated params) | Any2En | | | | | En2Any | | | | |
|---|---|---|---|---|---|---|---|---|---|---|
| | High | Medium | Low | Ex-Low | Avg | High | Medium | Low | Ex-Low | Avg |
| Baselines | | | | | | | | | | |
| Dense (0.4B / 0.4B) | 29.9 | 28.1 | 24.2 | 15.0 | 25.5 | 37.7 | 22.3 | 16.6 | 8.7 | 20.1 |
| MoE (3.5B / 0.5B) | 31.5 | 30.4 | 26.5 | 15.8 | 27.5 | 39.5 | 24.7 | 18.7 | 11.2 | 22.4 |
| mRASP2 (0.4B / 0.4B) | 30.0 | 30.7 | 26.1 | 17.5 | 27.7 | 40.3 | 25.6 | 19.8 | 11.2 | 23.2 |
| M2M-100 (1.2B / 1.2B) | 30.0 | 27.3 | 22.3 | 17.0 | 24.8 | 34.6 | 22.2 | 16.3 | 12.7 | 20.3 |
| NLLB (1.3B / 1.3B) | 32.3 | 33.1 | 31.2 | 28.7 | 31.9 | 36.3 | 26.6 | 21.7 | 19.8 | 24.6 |
| MMNMT | | | | | | | | | | |
| MMNMT (Dense) (3.5B / 0.5B) | 32.1 | 31.0 | 27.4 | 17.0 | 28.3 | 39.8 | 24.8 | 19.3 | 10.7 | 22.6 |
| MMNMT (mRASP2) (3.5B / 0.5B) | 32.9 | 32.1 | 28.8 | 18.8 | 29.5 | 41.0 | 26.1 | 20.4 | 12.3 | 23.9 |

Table 3: Results of the off-the-shelf dense-to-sparse adaptation experiments on the test sets of PC32.

| Model | Ar | De | Fr | Nl | Ru | Zh | Avg |
|---|---|---|---|---|---|---|---|
| MoE | 5.7(52.9%) | 5.6(25.4%) | 7.0(22.8%) | 5.2(17.4%) | 5.1(17.4%) | 3.2(8.0%) | 5.3(24.0%) |
| MMNMT | 12.0(80.9%) | 7.8(35.0%) | 10.2(36.9%) | 6.8(25.1%) | 11.7(49.4%) | 9.1(42.4%) | 9.6(44.9%) |

Table 4: Detailed results of zero-shot translation. The numbers outside the brackets are the BLEU scores, and the numbers in the brackets are the language accuracy.

spectively, where both the encoder and decoder of the adapted models were initialized by the counterpart modules from the corresponding models. As we can see in Table 3, mRASP2 is a very strong baseline and is consistently better than both dense and MoE baselines. MMNMT (Dense) achieves a consistent improvements compared with MoE baselines on Any-to-English translation (28.3 vs. 27.5). MMNMT (mRASP2) can further significantly improve translation performance over mRASP2, especially on Any-to-English translation (29.5 vs. 27.7). On English-to-Any translation, MMNMT (mRASP2) gains an average improvement of 0.7 BLEU and a substantial improvement of 1.1 BLEU on extremely low-resource language translation. These results suggest that our modularized MNMT framework can be successfully applied to off-the-shelf MNMT models to recycle and upgrade them. Our MMNMT model achieves comparable performance with NLLB-1.3B on low-resource language translation. NLLB-1.3B uses far more training data than ours (only 108K sentence pairs in total) on extremely low-resource languages. It is hence reasonable that our model cannot compete with it on these languages. Note that our framework can also use the modules of NLLB-1.3B to achieve further performance improvements.

## 6 Conclusion

In this paper, we have presented MMNMT, a modularized multilingual neural machine translation framework that is capable of flexibly assembling both dense and sparse blocks to achieve the best of both worlds for high/low-resource language translation and zero-shot translation. Experiments and in-depth analyses demonstrate that our framework combined with different modules significantly outperforms the MoE and dense baselines on high- and low-resource language translation as well as zero-shot translation.

## Limitations

The MMNMT model is able to incorporate different modules from different basic models to combine their strengths. However, these modules might be not consistent with each other, making them not able to fit together in a single model. We leave this to our future work for exploring new strategies.

## Acknowledgments

The present research was partially supported by the Key Research and Development Program of Yunnan Province (Grant No. 202203AA080004). We would like to thank the anonymous reviewers for their insightful comments.

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

## A  Languages Grouping Result

We built language-group-specific routers for MoE following section 4.1, the grouping results are shown in Table 5.

## B  The Experiment Setting for mRASP2 Adapting Experiments

**Dataset**   We downloaded the binary file of PC32 dataset without RAS and checkpoint from the project website[3]. We grouped these languages into four categories (excluding English-Mongolian language pair) according to their corpus size: High (10M $\leq$ size, 4 directions), Medium (1M $\leq$ size $<$ 10M, 32 directions), Low (100K $\leq$ size $<$ 1M, 16 directions), Extremely Low (size $<$ 100K, 10 directions). We omitted Maltese and Esperanto to make all multilingual translation models comparable in test. We report the validation and test set used in mRASP2 adapting experiments in Table 6.

**Model Configurations**   Our dense MNMT model contained 12 Transformer encoder blocks and 12 Transformer decoder blocks, where the model dimension was 1024, the number of attention heads was 16. The position embeddings of encoder and decoder were learned during training. For the MoE model, MoE layer frequency of the MoE MNMT model was 2, the capacity factor was 1.25 in training and the eval-capacity-factor was set to 0.75 in inference. Every MoE layer had 32 experts and Top-2 routing strategy was used for expert routing. Other configurations were the same as the dense model.

**Training Configurations**   We conducted all experiments on 8 NVIDIA Tesla V100 32GB GPUs. For dense model pre-training and MMNMT fine-tuning, we used the Adam optimizer (Kingma and Ba, 2014) with learning rate = $2e^{-4}$, $\alpha = 0.9, \beta = 0.98$. The learning schedule was inverse sqrt decay with the number of warm-up steps being set to 4000. The total number of updates was 100K. We set the training max tokens to 2000 per GPU and accumulated the gradients every 23 steps. The training objects were cross-entropy loss and load-balance loss with a weight of 0.1. The best checkpoint was selected according to the perplexity (ppl) on the validation set. The BLEU score was computed vias sacrebleu (Post, 2018).

---

[3]https://github.com/PANXiao1994/mRASP2

| Groups | Languages |
|---|---|
| 1 | as,az,bn,br,cy,et,fa,fi,ga,gd,gu,hi,hu,id,kk,kn,ku,ky,mg,ml,mr,ms,ne,or,pa,ps,se,si,ta,te,tg,tk,tr,tt,ug,ur,uz |
| 2 | am,ar,be,bg,bs,cs,ha,he,hr,ka,mk,mt,pl,ru,sh,sk,sl,sr,uk |
| 3 | af,ca,da,de,en,eo,es,fr,fy,gl,is,it,li,nb,nl,nn,no,oc,pt,ro,sv,wa,yi |
| 4 | el,eu,ig,ja,km,ko,lt,lv,my,rw,sq,th,vi,xh,zh,zu |

Table 5: Grouping results on the OPUS-100 dataset.

| Lang Pair | Validation set | Test set |
|---|---|---|
| en-fr | newstest13 | newstest14 |
| en-de | newstest13 | newstest14 |
| en-zh | newsdev17 | newstest17 |
| en-ro | newsdev16 | newstest16 |
| en-cs | newstest15 | newstest16 |
| en-tr | newsdev16 | newstest16 |
| en-ru | newstest18 | newstest19 |
| en-fi | newstest16 | newstest17 |
| en-es | newstest12 | newstest13 |
| en-it | newssyscomb2009 | newstest2009 |
| others | OPUS-100-valid | OPUS-100-test |

Table 6: The validation and test set used in mRASP2 adapting experiments