# OpenReview forum: "MMNMT: Modularizing Multilingual Neural Machine Translation with Flexibly Assembled MoE and Dense Blocks"
_EMNLP/2023/Conference — EMNLP 2023 Main_

### Official Review · Reviewer_E1x3 · 2023-08-04

**Soundness:** 4

**Excitement:**

3: Ambivalent: It has merits (e.g., it reports state-of-the-art results, the idea is nice), but there are key weaknesses (e.g., it describes incremental work), and it can significantly benefit from another round of revision. However, I won't object to accepting it if my co-reviewers champion it.

**Paper Topic And Main Contributions:**

This paper proposes an MNMT framework that combines the advantages of dense and MoE models to achieve the best of both worlds and also introduces a three-stage training method for the model.  Experimental results on two benchmarks show the effectiveness of the proposed model in which the model outperforms the dense and MoE.

**Reasons To Accept:**

The model architecture is interesting especially the training method for the model. The description of the training method is also clear and simple to implement. The evaluations of different tasks and results cover multi parts of the model and show the effectiveness.

**Reasons To Reject:**

However, there are some problems in this paper.

Authors have introduced some similar works in the related work while Section 5 only gives results of mRASP2. Authors should give more results of existing works for evaluation. The authors should also give the amount of parameters in Section 5.

Figure 1, Figure 4 and 5 are too small.

**Reproducibility:**

3: Could reproduce the results with some difficulty. The settings of parameters are underspecified or subjectively determined; the training/evaluation data are not widely available.

**Reviewer Confidence:**

3: Pretty sure, but there's a chance I missed something. Although I have a good feel for this area in general, I did not carefully check the paper's details, e.g., the math, experimental design, or novelty.

---

> ### Author Rebuttal · Authors · 2023-08-29
>
> Thank you so much for your insightful comments and valuable suggestions. Our answers to your questions and comments are as follows.
>
> Answer to RejectionReason1:
>
> We have recently conducted experiments against 2 additional strong baseline models for a more in-depth comparison, namely NLLB-1.3B[1] and M2M-100[2]. The new results are as follows:
> |            Model (#params / #activated params)             |      |        | Any2En |        |           |      |        | En2Any |        |      |
> |:---------------------------:|:----:|:------:|:------:|:------:|:---------:|:----:|:------:|:------:|:------:|:----:|
> |                             | High | Medium |   Low  | Ex-Low |    Avg    | High | Medium |   Low  | Ex-Low |  Avg |
> |                             |      |        |        |        | Baselines |      |        |        |        |      |
> |     Dense (0.55B / 0.55B)     | 29.9 |  28.1  |  24.2  |  15.0  |    25.5   | 37.7 |  22.3  |  16.6  |   8.7  | 20.1 |
> |       MoE (3.7B / 0.65B)      | 31.5 |  30.4  |  26.5  |  15.8  |    27.5   | 39.5 |  24.7  |  18.7  |  11.2  | 22.4 |
> |     mRASP2 (0.55B / 0.55B)    | 30.0 |  30.7  |  26.1  |  17.5  |    27.7   | 40.3 |  25.6  |  19.8  |  11.2  | 23.2 |
> |     M2M-100 (1.2B / 1.2B)     | 30.0 |  27.3  |  22.3  |  17.0  |    24.8   | 34.6 |  22.2  |  16.3  |  12.7  | 20.3 |
> |       NLLB (1.3B / 1.3B)      | 32.3 |  33.1  |  31.2  |  28.7  |    31.9   | 36.3 |  26.6  |  21.7  |  19.8  | 24.6 |
> |                             |      |        |        |        |   MMNMT   |      |        |        |        |      |
> |  MMNMT (Dense) (3.7B / 0.65B) | 32.1 |  31.0  |  27.4  |  17.0  |    28.3   | 39.8 |  24.8  |  19.3  |  10.7  | 22.6 |
> | MMNMT (mRASP2) (3.7B / 0.65B) | 32.9 |  32.1  |  28.8  |  18.8  |    29.5   | 41.0 |  26.1  |  20.4  |  12.3  | 23.9 |
>
> The numbers in bracket are the number of model parameters and activated parameters at inference time. It is worth noting that the newly compared models have larger model size, larger training data and longer training time than our models, while our method improves translation quality to a level comparable to them.
>
> Our MMNMT model achieves comparable performance with NLLB-1.3B on low-resource languages translation. NLLB has far more training data than ours (only 108K sentence pairs in total) on extremely low-resource languages. It is hence reasonable that our model cannot compete with it on these languages. Note that our framework can also use the modules of NLLB-1.3B to achieve further performance improvements. We’ll provide these results in the next version.
>
> We have provided the number of model parameters of each baseline and MMNMT in Section 5 and the number of parameters actually activated during inference in the above table. We will provide all these details and adjust the size of Figure 1, 4 and 5 according to your suggestion in the next version.
>
> [1] Costa-jussà M R, Cross J, Çelebi O, et al. No language left behind: Scaling human-centered machine translation[J]. arXiv preprint arXiv:2207.04672, 2022.
>
> [2] Fan A, Bhosale S, Schwenk H, et al. Beyond english-centric multilingual machine translation[J]. The Journal of Machine Learning Research, 2021, 22(1): 4839-4886.

---

### Official Review · Reviewer_bR2A · 2023-08-05

**Soundness:** 4

**Excitement:**

4: Strong: This paper deepens the understanding of some phenomenon or lowers the barriers to an existing research direction.

**Paper Topic And Main Contributions:**

This paper aims at improving multilingual machine translation. The authors propose modularized MNMT framework which is able to flexibly assemble dense and MoE-based sparse modules. Experiment results on OPUS-100 dataset show that the proposed technique can bring improvement against the MoE baseline. Overall, the authors make a good contribution on improving MoE framework.

**Reasons To Accept:**

1. interesting topic: mixture-of-experts (MoE) is currently a popular topic and the authors explore it in multilingual machine translation.
2. comprehensive analysis: the authors conduct extensive experiment and analysis to show the effectiveness of the proposed method.

**Reasons To Reject:**

1. missing comparison: this paper lacks comparison to strong multilingual machine translation baselines, e.g., NLLB[1], Lego-MT[2]. Can MMNMT outperforms these strong baselines? Is there a limitation on current MoE framework?

[1] No language left behind: Scaling human-centered machine translation.NLLB team. arXiv:2207.04672.

[2] Lego-MT: Towards detachable models in massively multilingual machine translation. Yuan et al. arXiv:2212.10551.

**Reproducibility:**

3: Could reproduce the results with some difficulty. The settings of parameters are underspecified or subjectively determined; the training/evaluation data are not widely available.

**Reviewer Confidence:**

3: Pretty sure, but there's a chance I missed something. Although I have a good feel for this area in general, I did not carefully check the paper's details, e.g., the math, experimental design, or novelty.

---

> ### Author Rebuttal · Authors · 2023-08-29
>
> Thank you so much for your insightful comments and valuable suggestions. Our answers to your questions and comments are as follows.
>
> Answer to RejectionReason1:
>
> We have recently conducted experiments against 2 additional strong baseline models for a more in-depth comparison, namely NLLB-1.3B[1] and M2M-100[2]. We haven’t compared our model with Lego-MT[3] because we could not reproduce it as its public released model files are partially missing. The new results are as follows:
>
> |            Model (#params / #activated params)             |      |        | Any2En |        |           |      |        | En2Any |        |      |
> |:---------------------------:|:----:|:------:|:------:|:------:|:---------:|:----:|:------:|:------:|:------:|:----:|
> |                             | High | Medium |   Low  | Ex-Low |    Avg    | High | Medium |   Low  | Ex-Low |  Avg |
> |                             |      |        |        |        | Baselines |      |        |        |        |      |
> |     Dense (0.55B / 0.55B)     | 29.9 |  28.1  |  24.2  |  15.0  |    25.5   | 37.7 |  22.3  |  16.6  |   8.7  | 20.1 |
> |       MoE (3.7B / 0.65B)      | 31.5 |  30.4  |  26.5  |  15.8  |    27.5   | 39.5 |  24.7  |  18.7  |  11.2  | 22.4 |
> |     mRASP2 (0.55B / 0.55B)    | 30.0 |  30.7  |  26.1  |  17.5  |    27.7   | 40.3 |  25.6  |  19.8  |  11.2  | 23.2 |
> |     M2M-100 (1.2B / 1.2B)     | 30.0 |  27.3  |  22.3  |  17.0  |    24.8   | 34.6 |  22.2  |  16.3  |  12.7  | 20.3 |
> |       NLLB (1.3B / 1.3B)      | 32.3 |  33.1  |  31.2  |  28.7  |    31.9   | 36.3 |  26.6  |  21.7  |  19.8  | 24.6 |
> |                             |      |        |        |        |   MMNMT   |      |        |        |        |      |
> |  MMNMT (Dense) (3.7B / 0.65B) | 32.1 |  31.0  |  27.4  |  17.0  |    28.3   | 39.8 |  24.8  |  19.3  |  10.7  | 22.6 |
> | MMNMT (mRASP2) (3.7B / 0.65B) | 32.9 |  32.1  |  28.8  |  18.8  |    29.5   | 41.0 |  26.1  |  20.4  |  12.3  | 23.9 |
>
> The numbers in bracket are the number of model parameters and activated parameters at inference time. It is worth noting that the newly compared models have larger model size, larger training data and longer training time than our models, while our method improves translation quality to a level comparable to them.
>
> Our MMNMT model achieves comparable performance with NLLB-1.3B on low-resource languages translation. NLLB has far more training data than ours (only 108K sentence pairs in total) on extremely low-resource languages. It is hence reasonable that our model cannot compete with it on these languages. Note that our framework can also use the modules of NLLB-1.3B to achieve further performance improvements. We’ll provide these results in the next version.
>
> A possible limitation of our MoE framework might be that modules from different basic models might be not consistent with each other, making them not able to fit together in a single model losslessly. We leave this to our future work for exploring new strategies.
>
> [1] Costa-jussà M R, Cross J, Çelebi O, et al. No language left behind: Scaling human-centered machine translation[J]. arXiv preprint arXiv:2207.04672, 2022.
>
> [2] Fan A, Bhosale S, Schwenk H, et al. Beyond english-centric multilingual machine translation[J]. The Journal of Machine Learning Research, 2021, 22(1): 4839-4886.
>
> [3] Yuan F, Lu Y, Zhu W H, et al. Lego-MT: Towards Detachable Models in Massively Multilingual Machine Translation[J]. arXiv preprint arXiv:2212.10551, 2022.

---

### Official Review · Reviewer_yF46 · 2023-08-06

**Soundness:** 3

**Excitement:**

4: Strong: This paper deepens the understanding of some phenomenon or lowers the barriers to an existing research direction.

**Paper Topic And Main Contributions:**

This paper to modularize multilingual NMT with assembled modules from both dense and sparse models, named MMNMT. Experiments prove the effectiveness of the proposed modules on high-resource, low-resource, and zero-shot translations.

**Questions For The Authors:**

1.	The authors should provide more details on comparison of the size of the models. Do the performance gains result from the larger size of the model?
2.	Can the proposed modularized structure generalize to other multilingual tasks (e.g., XNLI and XQuAD)?
3.	One of the main idea of this paper is to add language-specific modules to improve the multilingual performances, which is similar to the recent works (e.g. [1]). The authors should explain the differences to show the uniqueness of the proposed ideas.

**Reasons To Accept:**

1.	This paper is well-structured with pilot experiments to clear illustrate their motivations. And the main ideas of this paper are simple and easy to follow.
2.	Extensive experiments are conducted to prove the effectiveness of MMNMT and they well support the listed challenges.
3.	The proposed framework is proved to be flexibly adapted to the off-the-shelf models to support the recycling.

**Reasons To Reject:**

1.	The authors should include more sota baselines in MNMT to make the comparisons.

**Reproducibility:**

4: Could mostly reproduce the results, but there may be some variation because of sample variance or minor variations in their interpretation of the protocol or method.

**Reviewer Confidence:**

3: Pretty sure, but there's a chance I missed something. Although I have a good feel for this area in general, I did not carefully check the paper's details, e.g., the math, experimental design, or novelty.

---

> ### Author Rebuttal · Authors · 2023-08-29
>
> Thank you so much for your insightful comments and valuable suggestions. Our answers to your questions and comments are as follows.
>
> Answer to RejectionReason1:
>
> We have recently conducted experiments against 2 additional strong baseline models for a more in-depth comparison, namely NLLB-1.3B[1] and M2M-100[2]. The new results are as follows:
> |            Model (#params / #activated params)             |      |        | Any2En |        |           |      |        | En2Any |        |      |
> |:---------------------------:|:----:|:------:|:------:|:------:|:---------:|:----:|:------:|:------:|:------:|:----:|
> |                             | High | Medium |   Low  | Ex-Low |    Avg    | High | Medium |   Low  | Ex-Low |  Avg |
> |                             |      |        |        |        | Baselines |      |        |        |        |      |
> |     Dense (0.55B / 0.55B)     | 29.9 |  28.1  |  24.2  |  15.0  |    25.5   | 37.7 |  22.3  |  16.6  |   8.7  | 20.1 |
> |       MoE (3.7B / 0.65B)      | 31.5 |  30.4  |  26.5  |  15.8  |    27.5   | 39.5 |  24.7  |  18.7  |  11.2  | 22.4 |
> |     mRASP2 (0.55B / 0.55B)    | 30.0 |  30.7  |  26.1  |  17.5  |    27.7   | 40.3 |  25.6  |  19.8  |  11.2  | 23.2 |
> |     M2M-100 (1.2B / 1.2B)     | 30.0 |  27.3  |  22.3  |  17.0  |    24.8   | 34.6 |  22.2  |  16.3  |  12.7  | 20.3 |
> |       NLLB (1.3B / 1.3B)      | 32.3 |  33.1  |  31.2  |  28.7  |    31.9   | 36.3 |  26.6  |  21.7  |  19.8  | 24.6 |
> |                             |      |        |        |        |   MMNMT   |      |        |        |        |      |
> |  MMNMT (Dense) (3.7B / 0.65B) | 32.1 |  31.0  |  27.4  |  17.0  |    28.3   | 39.8 |  24.8  |  19.3  |  10.7  | 22.6 |
> | MMNMT (mRASP2) (3.7B / 0.65B) | 32.9 |  32.1  |  28.8  |  18.8  |    29.5   | 41.0 |  26.1  |  20.4  |  12.3  | 23.9 |
>
> The numbers in bracket are the number of model parameters and activated parameters at inference time. It is worth noting that the newly compared models have larger model size, larger training data and longer training time than our models, while our method improves translation quality to a level comparable to them.
>
> Our MMNMT model achieves comparable performance with NLLB-1.3B on low-resource languages translation. NLLB has far more training data than ours (only 108K sentence pairs in total) on extremely low-resource languages. It is hence reasonable that our model cannot compete with it on these languages. Note that our framework can also use the modules of NLLB-1.3B to achieve further performance improvements. We’ll provide these results in the next version.
>
> Answer to question 1:
> No, the performance gains are NOT from the larger size of the model. We compared MMNMT with the MoE model of the same size, and achieved a significant improvement on the performance of low-resource and high-resource language translation. In addition, although the MoE model is larger, the number of model parameters activated at inference time is much smaller than those of baselines (e.g., 0.65B vs 1.3B). Hence, the performance gains do not result from the larger size of the model. We will conduct more experiments on model size and provide results in the next version.
>
> Answer to question 2:
> Yes. Although our work currently focuses on the multilingual translation task, we believe that as the size of the model becomes larger, the problem of overfitting on low-resource languages is widespread. For this, our method can be effectively used to improve other multilingual tasks including XNLI and XQuAD. We’d like to extend our approach to these tasks in our future work.
>
> Answer to question 3:
> We add language-group-specific routers to route tokens from different language groups to different experts. Different from other methods which add language-specific modules (Could you please provide the paper title of the reference [1] in your question 3 ?), our method only imposes a soft constraint on routing. Tokens can still be routed to any appropriate experts, which avoids the conflict between different languages in a gentler way and will not damage the capacity of MoE models. Compared with other methods, our method only needs to add a very small number of model parameters (~1M). We’ll explain the differences from recent works as you suggested in the next version.
>
> [1] Costa-jussà M R, Cross J, Çelebi O, et al. No language left behind: Scaling human-centered machine translation[J]. arXiv preprint arXiv:2207.04672, 2022.
>
> [2] Fan A, Bhosale S, Schwenk H, et al. Beyond english-centric multilingual machine translation[J]. The Journal of Machine Learning Research, 2021, 22(1): 4839-4886.

---

### Meta-Review · Area_Chair_3Eww · 2023-09-21

**Recommendation:** 4

**Metareview:**

The work proposes a modular MMT framework that is able to use both dense and sparse MoE modules, which brings together the benefits of both sparse and dense MMT models. The authors do various experiments in different settings to compare to as baselines and compare across various language pairs and resource settings. The authors and reviewers had good discussion and reviewers are in consensus for accept, in particular the authors added additional critical baselines. The authors do note differences to existing work such as NLLB in terms of amount of data, which is a reasonable point in terms of performance.

---

### Decision · Program_Chairs · 2023-10-07

**Decision:**

Accept-Main

**Comment:**

The work proposes a modular MMT framework that is able to use both dense and sparse MoE modules, which brings together the benefits of both sparse and dense MMT models. The authors do various experiments in different settings to compare to as baselines and compare across various language pairs and resource settings. The authors and reviewers had good discussion and reviewers are in consensus for accept, in particular the authors added additional critical baselines. The authors do note differences to existing work such as NLLB in terms of amount of data, which is a reasonable point in terms of performance.